# Inhaled Allergy Diagnostics and Treatment in a Polluted Environment

**DOI:** 10.3390/ijms26135966

**Published:** 2025-06-21

**Authors:** Marcel Mazur, Ewa Czarnobilska

**Affiliations:** Department of Clinical and Environmental Allergology, Medical College, Jagiellonian University, Botaniczna St. 3, 31-501 Krakow, Poland; marcel.mazur@uj.edu.pl

**Keywords:** inhaled allergy, molecular diagnostics, air pollution

## Abstract

Allergic diseases have been increasing in prevalence over the last years. In a polluted environment, this problem can worsen and become more complex. Long-term exposure to air pollution can lead to the aggravation of allergic rhinitis (AR) and even to the development of seasonal asthma. Climate changes can accelerate and extend the pollination season. Research indicates that air pollution may modify the properties of pollen, making it more aggressive. Asymptomatic allergic people disclose their allergies in a polluted environment. A polluted environment complicates the diagnosis of seasonal allergies. The treatment might be more challenging as standard allergy medications may not be enough to control symptoms. The causal treatment of allergic rhinitis is specific allergen immunotherapy (AIT), which may prove less effective in people living in a polluted environment. The problem may lie in the proper evaluation for AIT as well as the assessment of its effectiveness. To date, the best way to confirm an allergy and qualify a patient for AIT seems to be molecular diagnostics. The question arises whether patients exposed to air pollution, which could potentially reduce the effectiveness of AIT, are still eligible for AIT. It is also debatable whether molecular diagnostics remain effective in such cases. Advancing precision medicine alongside environmental management represents a critical pathway toward reducing the growing global burden of allergic diseases.

## 1. Introduction

The impact of pollen on human health is particularly visible in allergic diseases, as exposure to pollen, recognized as the leading aeroallergen, triggers type I hypersensitivities, including allergic responses of the conjunctiva (conjunctivitis) and the mucosa of the respiratory system (rhinitis, asthma). Epidemiological data show how significant this problem is. Allergic rhinitis (AR) alone affects more than 400 million people worldwide, with prevalence rates between 10% and 30% among adults and more than 40% among children [1]. Previous research shows that aeroallergens contribute to approximately 63% of allergic rhinitis cases, with pollen accounting for over 92% of these cases [2].

Allergic diseases have increased in prevalence over the last years, which may in part be due to climate change [3]. The prevalence of pollen allergies in Europe was already estimated to have increased to 40% in 2007 [4].

The duration of exposure, intensity of exposure, and the allergenicity of the pollen have a large geographical and temporal variability, which results in differences in the prevalence of pollen-associated AR between locations and periods [5,6].

Allergic reactions to pollen are a serious socio-economic problem as they may lead to sleep disturbance, impaired mental well-being, and decreased quality of life, loss of productivity, or lower school performance for children, generating substantial healthcare costs. Additionally, it is believed that the majority of allergy patients (90%) either do not receive treatment or are inadequately treated, despite the availability of effective therapies for allergic diseases [7].

The growing problem of air pollution seems to be responsible for the increase in the occurrence of allergic rhinitis and the development of a more aggressive form of the disease as long-term exposure to air pollution is associated with heightened rhinitis severity [8]. It has been observed that both air pollution and pollen counts significantly influence the severity of allergic rhinitis and the development of seasonal asthma [9].

Experimental studies have demonstrated that various environmental pollutants, including air pollutants and chemical substances, can worsen different types of allergies. Diesel exhaust particles (DEPs), composed of a mixture of particles and numerous chemical substances, exacerbate asthma, primarily due to the organic chemical components found in DEPs [10]. Gene polymorphisms may affect an individual’s susceptibility to allergens in a polluted environment and their immune response as the GSTM1 null genotype was found to be associated with a lack of detoxifying enzyme activity and increased airway inflammation when exposed to DEP [11]. Mechanistically, this interplay is mediated through alterations in oxidative stress responses, epithelial barrier integrity, innate immune recognition, and adaptive immune signaling. Pollution exposure can also trigger epigenetic changes, like methylation, which can alter gene expression and further influence an individual’s sensitivity to pollutants [12]. Furthermore, it has become clear that ultrafine nanomaterials and Asian sand dust particles can intensify allergic inflammation [10].

A polluted environment complicates the diagnosis of seasonal allergies, as the symptoms can resemble those of other illnesses, such as viral respiratory infections which develop more often in polluted air. Furthermore, treatment might be more challenging because allergies can be aggravated by pollution, and standard allergy medications may not be enough to control symptoms [13].

Despite the high effectiveness of immunotherapy, which is the preferred type of intervention in the absence of contraindications as confirmed by studies, some patients do not respond to this type of treatment (non-responders), or its effects do not meet the expectations placed on it (poor responders) [14]. There is also a question of whether exposure to air pollution may be responsible for the deterioration of the effectiveness of specific immunotherapies.

## 2. Changes in Pollen Phenomenology

The impacts of global climate change, in conjunction with environmental factors such as air pollution, urbanization, and microclimatic variability, are significantly exacerbating pollen-mediated allergic diseases. These environmental shifts are contributing to increased pollen production by wind-pollinated plants, advancing and prolonging pollen seasons, and thereby reducing the predictability and manageability of allergic responses (Figure 1). Moreover, augmented pollen allergenicity is associated with more severe health outcomes among sensitized individuals. The introduction and spread of invasive allergenic plant species are further expanding the spectrum of allergenic exposure, resulting in novel sensitizations [3].

There is now evidence for the synergistic effects of heat and air pollution and limited evidence for the synergistic effects of simultaneous exposure to [3] air pollution, pollen, and heat, as well as [2] air pollution and pollen on respiratory diseases [15,16]. Notably, children and adolescents are more sensitive to air pollution than adults [17].

Warmer temperatures and higher atmospheric carbon dioxide levels caused by climate change alter pollination seasons worldwide, prolong pollen seasons, and increase the pollen count in the air in some parts of the world [18,19]. There is an increasing trend in the annual amount of airborne pollen for many taxa in Europe, which is more pronounced in urban areas than semi-rural/rural areas [17]. Birch trees in several European locations started to show a trend toward earlier flowering and pollen release more than 20 years ago, which was correlated with warming trends [20,21].

Air pollutants can directly affect the properties of aeroallergens [22]. Exposure of oak pollen to higher levels of SO_2_ or NO_2_ significantly enhanced its fragility and caused disruption, resulting in an increased release of pollen cytoplasmic granules [23]. Although pollution shows a minor effect on the physiological condition of B. pendula specimens, the Bet v1 (the main component responsible for the development of respiratory allergies) concentration measured in pollen samples collected in one of the most polluted cities in Europe was significantly higher than in places less polluted [24].

Although climate change may contribute to these changes, European researchers suggest that the anthropogenic rise in atmospheric CO_2_ levels, rather than increased temperatures, is a major influencing factor [19]. However, in animal models, it is the exposure to high temperatures that boosts the expression of inflammatory cytokines, induces oxidative stress in tissues, worsens airway hyperresponsiveness, and aggravates allergic asthma [25].

## 3. AIT as a Causal Method of Treatment

Allergen immunotherapy (AIT) is an efficacious and disease-modifying treatment option for IgE-mediated diseases, with AR and asthma among them. AIT may lead to clinical immunotolerance for years after the treatment completion [26]. AIT-induced tolerance involves a shift from a Th2 to a Th1 response, an increase in regulatory T and B cells, downregulation of proinflammatory effector cells, and the suppression of IgE. Additionally, AIT enhances the production of blocking antibodies IgG4, IgA, and IgD. It may also lead to a reduction in group 2 innate lymphoid cells (ILCs) while increasing ILC1 and ILC3 populations [27].

AIT is particularly beneficial for individuals who do not achieve adequate symptom control with medications or who experience side effects from pharmacological treatments [28]. There are two main types of allergen immunotherapy: subcutaneous immunotherapy (SCIT) and sublingual immunotherapy (SLIT). SCIT involves administering allergens through injections under the skin, while SLIT delivers allergen extracts in the form of drops or tablets placed under the tongue [28]. Both approaches have shown effectiveness and are suitable for different patient groups. Unlike symptomatic treatments, AIT helps to alter the long-term progression of allergic diseases [29].

Climate change is anticipated to have a substantial impact on the prescribing practices for SLIT. Projected extensions in pollen seasons may necessitate earlier initiation of the therapy and prolonged treatment durations. Moreover, the anticipated rise in cumulative pollen exposure days could theoretically elevate the risk of adverse events associated with SLIT [30]. In view of the exacerbation of allergic inflammatory responses observed during episodes of heightened ambient air pollution, it would be worth considering lowering initial doses of AIT to reduce adverse reaction risks, especially during pollution peaks and a slower titration schedule compared with standard protocols.

The basis for the success of AIT is a proper diagnosis and evaluation. Due to the nonspecific nature of the main symptoms of AR and asthma (runny nose, cough, and dyspnea), the frequent lack of changes in physical examinations, and the normal results of additional tests in asthma, apart from periods of exacerbations, a differential diagnosis may be required.

## 4. Diagnostic Methods Used in Qualification for AIT

Allergy diagnostics rely on two types of tests: in vivo (skin tests and provocations) and in vitro (detecting specific IgE). The most commonly used diagnostic method is the skin prick test (SPT). Despite its numerous advantages, its sensitivity and specificity are affected by factors such as the quality of the extract, its biological potency, and the allergen concentration [31].

An alternative to skin tests is in vitro testing. All available methods are based on detecting allergen-specific IgE antibodies (sIgE). The most commonly used tests are ELISA (enzyme-linked immunosorbent assay), ImmunoCAP, ImmunoCAP ISAC, or Alex Polycheck. In vitro tests have become the basis for component-resolved diagnosis (CRD). Testing individual allergens provides a wealth of information—apart from identifying which allergen causes an allergic reaction, it also helps determine potential cross-reactivity and the risk of severe allergic reactions. Unfortunately, the methodology is not without its flaws. To obtain reliable results, standardized procedures for interpreting the results are necessary. Moreover, the tested allergens must have established significance in the disease process, and their quality (tertiary structure despite cross-linking on the matrix) must be satisfactory [32]. Analyzing the sensitivity and specificity of SPT, it is worth comparing them with studies based on sIgE, which may show divergent values. Due to the way epitopes are recognized—IgE binds to linear epitopes (derived from the amino acid sequence, i.e., the primary structure of the protein), as well as conformational (structural) epitopes—both the primary and tertiary structures seem to be crucial elements of the quality of allergens used in diagnostics [33]. In addition to structure, the glycosylation profile of a protein is also a determining factor in its allergenicity [34].

It is commonly acknowledged that the benefits of skin testing include its use of testing on an end organ, lower cost, high sensitivity, and the provision of immediate results that are easily visible to the patient [35]. In a polluted setting, SPT remained superior in specificity and predictive accuracy, while sIgE panels offered higher sensitivity [36]. However, in patients with negative or ambiguous skin test results, in vitro tests are used. From a clinical perspective they are useful for identifying cross-reactivity between pollens of the same as well as of a different group.

A study comparing different diagnostic methods for inhaled allergies evaluated the accuracy and information provided by two microarray platforms (Microtest and ISAC) alongside standard diagnostic methods (ImmunoCAP and skin prick test) in a cohort of 71 children with severe and controlled asthma. The study found that the microarray methods offered additional insights in 47% of sensitized children compared to standard methods [37].

The importance of CRD in accurately identifying disease-eliciting molecules, which is crucial for effective immunotherapy, was highlighted in patients with respiratory allergic diseases. This research compared sensitization profiles determined by skin prick tests and CRD, allowing for a more precise description of the sensitization profile and helping to identify connections between symptoms and specific antigens, suggesting that patients with respiratory allergies may benefit from the routine use of CRD [38].

Additionally, there is research providing insights into the efficacy of various diagnostic methods, including skin-prick, intradermal, and serum-specific IgE tests, in diagnosing environmental allergies [39].

These studies collectively underscore the significance of employing multiple diagnostic methods, including molecular techniques like CRD, to enhance the accuracy of inhaled allergy diagnoses.

Common diagnostic techniques for allergies evaluate the presence or absence of allergen-specific sensitization (SPT or specific IgE results), clearly identifying the key allergen(s) causing the AR and/or asthma symptoms, but none fully correlates with clinical outcomes [40] (Table 1). CRD is a newer, more accurate method for identifying the specific allergen responsible. Although CRD may have a role in deciding which aeroallergen(s) should be chosen for AIT, definitive trials are awaited. It is a matter of debate whether skin testing, IgE determination including CRD, or the basophil activation test (BAT) are able to distinguish between asymptomatic sensitization and clinically relevant allergies [40]. BAT, which is a flow cytometry-based diagnostic assay that quantifies the expression of activation markers, such as CD63, on the surface of basophils following in vitro stimulation with a specific allergen, may have an advantage here. In comparison with in vivo provocation tests, BAT offers greater convenience and enhanced patient safety, as it necessitates only a peripheral blood sample rather than direct allergen exposure in the patient, but it is poorly available and expensive [41].

As a result, it is important to determine whether patients are: (i) polysensitized and clinically mono-allergic (with only one allergen causing their symptoms), or (ii) polysensitized and poly-allergic (experiencing symptoms due to exposure to multiple allergens) [40]. This distinction may be evident from the patient’s history or may require further investigation using component-resolved diagnostics or nasal/conjunctival provocation challenges, provided the clinician is experienced in these diagnostic techniques [42].

Moreover, the reason for the suboptimal effects of immunotherapy may be a misclassification resulting from the clinical similarity between different endotypes of allergic rhinitis and asthma [7]. Allergic rhinitis and non-IgE-dependent, non-allergic eosinophilic rhinitis (NARES) may have a similar clinical picture and are characterized by the same biomarkers (eosinophilia, LTC4, ECP) [43]. In this way, a patient with NARES who has a concomitant clinically insignificant sensitization (bystander) may be qualified for immunotherapy, from which he or she will not benefit [14].

In addition, selecting the wrong allergen, which does not play a significant role in the etiology of allergy symptoms, results in the failure of AIT [14]. There are also differences in the response to AIT depending on which allergen the patient is being desensitized to [44].

In some cases, it should be assumed that the symptoms are not caused by an IgE-mediated allergy, i.e., tree or grass pollen, but instead by exposure to polluted air. As it is known, air pollution (particulate matter, sulfur and nitrogen oxides, ozone) has an impact on the symptoms of nonallergic rhinitis (NAR) [45,46,47]. There is also strong evidence linking exposure to air pollutants with an increased risk of allergic disorders [16]. Particulate matter (PM) 2.5 can activate peripheral blood basophils and induce the expression of CD63 receptors on their surface in both atopic and non-atopic individuals, while exposure to birch pollen and PM2.5 has a synergistic effect in sensitized individuals [48].

Experimental studies indicate that co-exposure to aeroallergens and DEPs enhances T-helper 2 (Th2)-mediated immune responses, potentially counteracting the immunomodulatory effects of AIT and reducing its clinical benefits [5]. Moreover, patients with persistent allergic rhinitis or polysensitization—conditions more prevalent in polluted urban settings—are more likely to exhibit suboptimal responses or non-responsiveness to AIT [44].

Atopic diseases are influenced by environmental factors that interact with an individual’s genetic predisposition to shape the risk of developing this disease [49]. When migrating to Western countries, individuals developing atopic diseases exhibit varying degrees of severity depending on their duration of residence or the timing of their migration [50]. Individuals with AR experience nasal hyperreactivity, making them more sensitive to airborne irritants. As a result, these irritants likely contribute to worsening AR symptoms. Air pollution can enhance the allergic immune response through their irritant effects. Additionally, pollution may harm the nasal mucosa and disrupt mucociliary clearance, which could allow inhaled allergens easier access to the nasal mucosal cells [49].

## 5. Approaches to Improving Diagnostics and Treatment Assessment

Precise monitoring and the prediction of trends in pollen concentration in the air using a machine deep learning method may be helpful in assessing the relationship between disease symptoms and exposure to plant pollen [51].

There is definitely a need for thorough clinical selection of patients who may benefit from AIT as some of them do not respond well to this therapy (Figure 2).

Biomedical big data can generally be categorized into two main types: omics and non-omics data, which both play crucial roles in understanding and interpreting allergic diseases. Non-omics data encompass laboratory test results, imaging and morphological parameters, environmental biomonitoring, mobile health (mHealth) records, and clinical registries maintained by healthcare professionals. In contrast, omics data are generated via high-throughput biological technologies and provide extensive features that reflect biological activity at various levels. These include genomics, epigenomics, metabolomics, transcriptomics, and proteomics [52].

To date, there has been no definite biomarker of the clinical evaluation of patients, although new options, like IgE–FAB, which is the inhibition of allergen binding to IgE evaluated by a validated flow cytometry-based and enzyme-linked immunosorbent-facilitated antigen binding assay (ELIFAB), arise [24]. Still, considerable advancement has to be achieved in the application of more precise phenotyping and endotyping based on advanced data management provided by mHealth records, CRD, and immune biomarker assessments [52]. A 2024 scoping review evaluated 48 general and allergy-specific mHealth apps. The study identified 14 key quality dimensions—including validity, safety, interoperability, usability, security, and clinical value—but noted that few apps met more than one or two criteria, stressing the need for structured quality assurance systems [53].

As of today, the BAT may serve as a method that can assess sensitizations to both pollen and particulate matter, helping to better select candidates for AIT [46].

Finally, it should not be forgotten that individuals with inhaled allergies should take measures to reduce exposure to allergens and pollutants, such as avoiding going outside during peak pollen hours, using air purifiers indoors, and wearing protective masks. They should also pay attention to the pollen forecast that takes into account the influence of weather conditions. Combining forecasts with symptom diaries and expert counseling supports daily decisions—like staying indoors, planning activities, or initiating medication pre-seasonally. Patients often trivialize symptoms and feel that self-managing is easier than consulting doctors. Interventions must build trust, highlight condition impacts, and include interactive tools [54]. Models including pollen-forecast-assisted action plans and clear behavioral prompts in the user interface improve engagement [55]. A silver lining for all those struggling with the problem of air pollution could be the fact that improved air quality is associated with a reduction in the prevalence of allergic diseases in children, as confirmed by both European and Asian studies [56,57]. Therefore, all possible actions aimed at improving air quality should be taken both at the local and global levels.

## 6. Conclusions

The rising prevalence and clinical severity of pollen-mediated allergic diseases reflect a complex and multifactorial interplay between environmental change and individual susceptibility. Climatic factors—particularly global warming, elevated atmospheric carbon dioxide levels, and microclimatic variability—along with increasing air pollution and urbanization, contribute to prolonged pollen seasons, elevated pollen concentrations, and enhanced allergenicity. These phenomena, in turn, exacerbate the burden of allergic diseases, especially in vulnerable populations such as children and urban residents.

AIT remains the only disease-modifying intervention for IgE-mediated allergic conditions, offering the potential for sustained clinical benefits beyond treatment cessation. However, its efficacy is critically dependent on accurate patient selection, precise identification of clinically relevant allergens, and the use of validated diagnostic methodologies. Despite advances in molecular allergology, limitations remain in distinguishing asymptomatic sensitization from clinical manifestations of allergy, particularly in polysensitized individuals or in the presence of overlapping environmental exposures.

Environmental pollution not only amplifies allergic inflammation but may also reduce the efficacy of AIT, thereby underscoring the necessity for integrated approaches to allergy management that consider both immunological and environmental determinants. Future efforts should prioritize the improvement of phenotyping and endotyping techniques, development of reliable clinical biomarkers, and adoption of emerging technologies such as mobile health (mHealth) tools and machine learning for exposure prediction.

Moreover, the observed association between air quality and allergic morbidity in children further reinforces the need for coordinated action aimed at reducing air pollution. In this context, advancing precision medicine alongside environmental management represents a critical pathway toward reducing the growing global burden of allergic diseases.

## Figures and Tables

**Figure 1 ijms-26-05966-f001:**
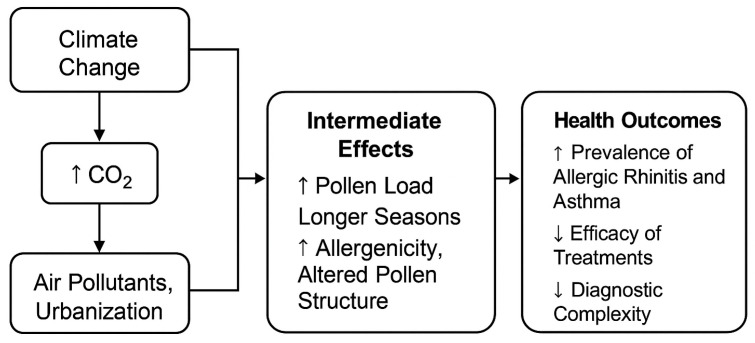
Schematic Representation of Environmental Impacts on Pollen and Allergy Burden.

**Figure 2 ijms-26-05966-f002:**
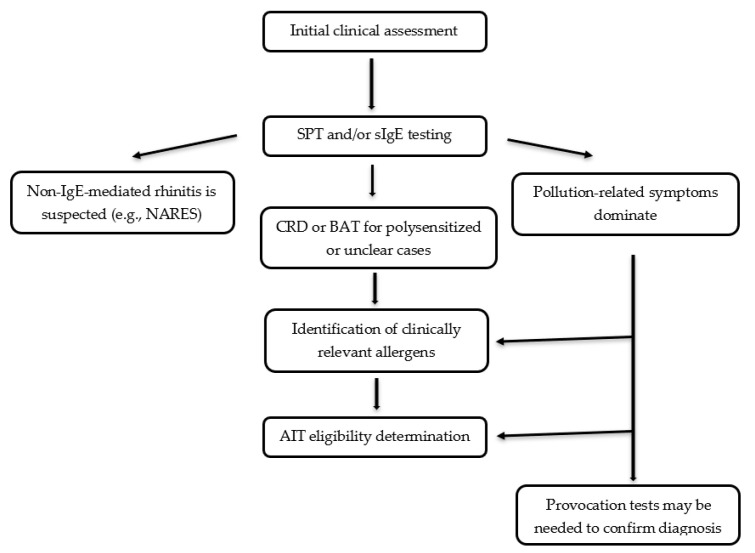
Overview of Diagnostic Workflow for AIT Qualification.

**Table 1 ijms-26-05966-t001:** Comparison of Diagnostic Tools for Aeroallergen Sensitization.

Diagnostic Method	Type	Advantages	Limitations	Clinical Value
Skin Prick Test (SPT)	In vivo	Low cost, quick, end-organ testing, immediate results	Sensitivity affected by extract quality, false negatives possible	First-line for most suspected allergies
Specific IgE (ImmunoCAP)	In vitro	Quantitative, no risk of systemic reaction	Costly, may not reflect clinical relevance	Useful when skin test is contraindicated
Component-Resolved Diagnosis (CRD)- Microarray (ISAC, Alex)	In vitro	Identifies specific protein components, detects cross-reactivity, provides extensive allergen profile, useful for polysensitized patients	Expensive, interpretation requires expertise, lower sensitivity for low sIgE levels	Increasing use for AIT qualification, supplementary to routine tests
Basophil Activation Test (BAT)	In vitro	Reflects functional sensitization, identifies response to both pollen and pollution	Expensive, not widely available	Emerging tool for patient stratification and AIT response
Provocation Tests	In vivo	Mimics natural exposure, confirms clinical relevance	Time-consuming, risk of reaction	Confirmatory in complex or ambiguous cases

## Data Availability

No new data were created or analyzed in this study. Data sharing is not applicable to this article.

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
