# Peer review of "Inhaled Allergy Diagnostics and Treatment in a Polluted Environment"

_ijms, 2025, doi:10.3390/ijms26135966_

Round 1
Reviewer 1 Report
Comments and Suggestions for Authors
This review conducts a relatively comprehensive exploration of the diagnosis of inhaled allergies in a polluted environment. However, there are some major issues, including research methods, research contents, and clinical applications.
- Research Methods.
In the evaluation of diagnostic methods, although the advantages and disadvantages of various diagnostic methods are mentioned, such as the skin prick test being affected by factors like the quality of the extract and the need for standardized procedures in in - vitro testing methods. However there is a lack of direct comparative research data on the diagnostic accuracy of these methods in a polluted environment. There is no specific quantification of the differences in the probabilities of misdiagnosis and missed diagnosis of different diagnostic methods under the interference of a polluted environment, making it difficult to accurately judge the reliability of each method in actual clinical applications.
The introduction of new diagnostic technologies such as the machine - learning - based method for predicting pollen concentration remains at a theoretical level. There is no mention of its practical application effects, accuracy verification, or the evaluation of its combined use with traditional diagnostic methods. Tables and Figure s are required for readers to understand the content.
- Incomplete summary of research summary.
When exploring the impact of air pollution on allergy diagnosis and treatment, the research mainly focuses on the research results and data from European regions. However, there are differences in air pollution components, pollen types, and the allergic constitution of the population in different regions. The lack of comparative studies in multiple regions around the world limits the universality of the research conclusions.
The paper does not deeply explore the interaction mechanism between genetic factors and environmental factors (especially air pollution) in the development of allergies. Although it is mentioned that atopic diseases are jointly affected by genetics and the environment, it does not elaborate in detail how gene polymorphisms affect an individual's susceptibility to allergens in a polluted environment and the immune response. The incomplete summaries make it difficult to comprehensively explain the complex mechanisms of the occurrence and development of allergic diseases.
- Insufficient guidance for clinical application.
Regarding the problem that the effectiveness of immunotherapy is affected by air pollution, only issues related to evaluation and effectiveness are raised, but specific suggestions for optimizing the immunotherapy plan in a polluted environment are not provided. For example, it is not clear whether it is necessary to adjust the dosage and course of immunotherapy or combine other treatment methods to improve the curative effect, which lacks practical guiding significance for clinical practice.
In terms of patient management, although it is mentioned that patients should take measures to reduce exposure to allergens and pollutants, specific implementation strategies and supervision methods are not provided. There is no further discussion on how to guide patients' daily lives based on pollen concentration prediction results and how to ensure that patients effectively implement protective measures, which is not conducive to patients' management and disease control.
Author Response
On behalf of my co-author and myself, I would like to thank you for your comments. We have made every effort to address all concerns raised by the reviewers — we introduced revisions, clarifications, and new sections of text, as well as added a table and two diagrams. We have also refined imprecise wording in order to improve the overall language quality of the manuscript.
- Research Methods.
In the evaluation of diagnostic methods, although the advantages and disadvantages of various diagnostic methods are mentioned, such as the skin prick test being affected by factors like the quality of the extract and the need for standardized procedures in in - vitro testing methods. However there is a lack of direct comparative research data on the diagnostic accuracy of these methods in a polluted environment. There is no specific quantification of the differences in the probabilities of misdiagnosis and missed diagnosis of different diagnostic methods under the interference of a polluted environment, making it difficult to accurately judge the reliability of each method in actual clinical applications.
That sections have been provided.
The introduction of new diagnostic technologies such as the machine - learning - based method for predicting pollen concentration remains at a theoretical level. There is no mention of its practical application effects, accuracy verification, or the evaluation of its combined use with traditional diagnostic methods. Tables and Figure s are required for readers to understand the content.
The necessary section has been provided.
- Incomplete summary of research summary.
When exploring the impact of air pollution on allergy diagnosis and treatment, the research mainly focuses on the research results and data from European regions. However, there are differences in air pollution components, pollen types, and the allergic constitution of the population in different regions. The lack of comparative studies in multiple regions around the world limits the universality of the research conclusions.
The paper does not deeply explore the interaction mechanism between genetic factors and environmental factors (especially air pollution) in the development of allergies. Although it is mentioned that atopic diseases are jointly affected by genetics and the environment, it does not elaborate in detail how gene polymorphisms affect an individual's susceptibility to allergens in a polluted environment and the immune response.
The incomplete summaries make it difficult to comprehensively explain the complex mechanisms of the occurrence and development of allergic diseases.
The suggested lacking section has bee implemented.
- Insufficient guidance for clinical application.
Regarding the problem that the effectiveness of immunotherapy is affected by air pollution, only issues related to evaluation and effectiveness are raised, but specific suggestions for optimizing the immunotherapy plan in a polluted environment are not provided. For example, it is not clear whether it is necessary to adjust the dosage and course of immunotherapy or combine other treatment methods to improve the curative effect, which lacks practical guiding significance for clinical practice.
The addressing paragraph has been added to the text.
In terms of patient management, although it is mentioned that patients should take measures to reduce exposure to allergens and pollutants, specific implementation strategies and supervision methods are not provided. There is no further discussion on how to guide patients' daily lives based on pollen concentration prediction results and how to ensure that patients effectively implement protective measures, which is not conducive to patients' management and disease control.
That section has also been supplemented.
We have introduced revisions, clarifications, and new sections of text, as well as added a table and two diagrams. We have also refined imprecise wording in order to improve the overall language quality of the manuscript.
We hope that the changes we have made will be positively received and will allow for the publication of our work.
Sincerely,
Marcel Mazur
Reviewer 2 Report
Comments and Suggestions for Authors
To the authors:
- General comments:
The review article entitled “Inhaled allergy diagnostics in a polluted environment” is revision that aims to discuss the current strategies used in AIT and how pollution hamper the diagnosis and increase the severity of the symptoms. The manuscript is interesting, however, I consider that the work has some major comments that authors should comment or change before its publications:
- Specific comments for revision: b) major.
- Please adapt more the title of the review to the content, as not only diagnosis of respiratory allergy is covered.
- Lines 164-170. Seems to lack references, and also some context, please check the sentence.
- Lines 198-205. Include a paragraph discussing if AIT is hampered by air pollution, and if in polluted cities there is a higher percentage of non-responders or poor responders of AIT. This is important as is part of the aim of the review.
- Line 226. Please revise the sentence, what do you mean by mHealth?
- Section 5. I strongly recommend going deeper on the machine learning and omics.
- I strongly suggest including a conclusion on the reviewed topic.
Minor comments:
- Line 103. AIT must be first defined in the text before using the abbreviation.
- Line 133. In vitro goes in italics.
- Line 133, 174. CRD has been abbreviated.
There are some text that lacks context, for example:
- Lines 164-170. Seems to lack references, and also some context, please check the sentence.
- Line 226. Please revise the sentence, what do you mean by mHealt?
Author Response
On behalf of my co-author and myself, I would like to thank you for your comments. We have made every effort to address all concerns raised by the reviewers — we introduced revisions, clarifications, and new sections of text, as well as added a table and two diagrams. We have also refined imprecise wording in order to improve the overall language quality of the manuscript.
The review article entitled “Inhaled allergy diagnostics in a polluted environment” is revision that aims to discuss the current strategies used in AIT and how pollution hamper the diagnosis and increase the severity of the symptoms. The manuscript is interesting, however, I consider that the work has some major comments that authors should comment or change before its publications:
- Specific comments for revision: b) major.
- Please adapt more the title of the review to the content, as not only diagnosis of respiratory allergy is covered. The title has been adapted.
- Lines 164-170. Seems to lack references, and also some context, please check the sentence. Refrerence has been provided.
- Lines 198-205. Include a paragraph discussing if AIT is hampered by air pollution, and if in polluted cities there is a higher percentage of non-responders or poor responders of AIT. This is important as is part of the aim of the review. A dedicated paragraph has been included.
- Line 226. Please revise the sentence, what do you mean by mHealth? Explained in the text.
- Section 5. I strongly recommend going deeper on the machine learning and omics. The fragment has been expanded.
- I strongly suggest including a conclusion on the reviewed topic.
Minor comments: All the comments mentioned below have been addressed.
- Line 103. AIT must be first defined in the text before using the abbreviation.
- Line 133. In vitro goes in italics.
- Line 133, 174. CRD has been abbreviated.
We hope that the changes we have made will be positively received and will allow for the publication of our work.
Sincerely,
Marcel Mazur
Reviewer 3 Report
Comments and Suggestions for Authors
Line 24: " are being properly evaluated for AIT". This passage is unclear: exactly what are the Authors trying to convene? That pollution somehow changes enrolment criteria for AIT? Consider rephrasing.
Lines 72-78: the entire subsection is extremely convoluted, consider rephrasing to better frame the effects of global climate change on pollen-mediated allergies.
Line 120: ANN? Please declare the acronym.
Line 141: "divergent values", as in? What do the Authors mean by "values"? The following passages
Line 148: "higher sensitivity" compared to?
Line 160: "skin prick tests and CRD with CRD", I assume the repetition is a typo
Lines 164-167: the entire paragraph is incorrectly formatted. Why was the paper declared by title rather than by a reference number?
Lines 179-180: consider elucidating how BAT works and how it could possibly be superior to other diagnostic techniques in discerning clinically significant allergy.
While the Author's work is commendable, the paper as a whole comes as a somewhat unorganized presentation of data: the title itself states "Inhaled allergy diagnostics in a polluted environment", but the main body of work tends to either present each facet as a disconnected, single-standing aspect of diagnostics or doesn't adequately make a point for how said techniques could/should be applied to pollution in allergic diseases. Furthermore, great attention has been given to AIT, of which there is no trace in the paper's title.
Comments on the Quality of English LanguageThe paper seems to suffer from syntax errors and convoluted passages, not to mention certain sentences that are not formatted according to scientific and AMA standards. Please consider an overhaul of the article.
Author Response
On behalf of my co-author and myself, I would like to thank you for your comments. We have made every effort to address all concerns raised by the reviewers — we introduced revisions, clarifications, and new sections of text, as well as added a table and two diagrams. We have also refined imprecise wording in order to improve the overall language quality of the manuscript.
Line 24: " are being properly evaluated for AIT". This passage is unclear: exactly what are the Authors trying to convene? That pollution somehow changes enrolment criteria for AIT? Consider rephrasing.
The passage has been rephrased.
Lines 72-78: the entire subsection is extremely convoluted, consider rephrasing to better frame the effects of global climate change on pollen-mediated allergies.
The subsection has been rephrased.
Line 120: ANN? Please declare the acronym.
The incorrectly used abbreviation has been changed.
Line 141: "divergent values", as in? What do the Authors mean by "values"? The following passages
A correction has been introduced.
Line 148: "higher sensitivity" compared to?
A correction has been introduced.
Line 160: "skin prick tests and CRD with CRD", I assume the repetition is a typo
A correction has been introduced.
Lines 164-167: the entire paragraph is incorrectly formatted. Why was the paper declared by title rather than by a reference number?
The reference number has been provided.
Lines 179-180: consider elucidating how BAT works and how it could possibly be superior to other diagnostic techniques in discerning clinically significant allergy.
BAT method has been elucidated.
While the Author's work is commendable, the paper as a whole comes as a somewhat unorganized presentation of data: the title itself states "Inhaled allergy diagnostics in a polluted environment", but the main body of work tends to either present each facet as a disconnected, single-standing aspect of diagnostics or doesn't adequately make a point for how said techniques could/should be applied to pollution in allergic diseases. Furthermore, great attention has been given to AIT, of which there is no trace in the paper's title.
The title of the paper has been adapted to better represent the body of the manuscript.
We hope that the changes we have made will be positively received and will allow for the publication of our work.
Sincerely,
Marcel Mazur
Round 2
Reviewer 1 Report
Comments and Suggestions for Authors
- An in-depth exploration of the interaction between gene polymorphisms and environmental pollution in allergic reactions.
The authors did not clarify whether they supplemented mechanistic analyses of this interaction.
- Optimization of Immunotherapy
The authors did not specify whether they provided actionable clinical guidelines for adjusting allergen immunotherapy (AIT) protocols (e.g., dosage, treatment duration) in polluted environments.
Author Response
- An in-depth exploration of the interaction between gene polymorphisms and environmental pollution in allergic reactions. The authors did not clarify whether they supplemented mechanistic analyses of this interaction. Response:
We have added a passage summarizing interaction between gene polymorphisms and environmental pollution.
- The authors did not specify whether they provided actionable clinical guidelines for adjusting allergen immunotherapy (AIT) protocols (e.g., dosage, treatment duration) in polluted environments. Response:
We have not been able to provide actionable clinical guidelines for adjusting allergen immunotherapy (AIT) protocols (e.g., dosage, treatment duration) in polluted environments as there are currently no clear, official guidelines in this area. However, we have proposed recommendations for action in this area and it has been included in the current, revised version of the manuscript.
Reviewer 3 Report
Comments and Suggestions for Authors
We'd like to thank the Authors for their efforts, which satisfy the previously raised points of concern
Author Response
1. We'd like to thank the Authors for their efforts, which satisfy the previously raised points of concern. Response: Dear Reviewer, thank you for your time and kind remarks, which have contributed to enhancing the quality of the article.